# Stakeholders' perceptions of the trends in contraceptive prevalence rate and total fertility rate in Ghana

**Fred Yao Gbagbo**[1]*, **Edward Kwabena Ameyaw**[2]

**1** Department of Health Administration and Education, Faculty of Science Education, University of Education, Winneba, Ghana, **2** Institute of Policy Studies and School of Graduate Studies, Lingnan University, Lingnan, Hong Kong

* gbagbofredyao2002@yahoo.co.uk, fygbagbo@uew.edu.gh

**Data Availability Statement:** Data cannot be shared publicly because of strict anonymity and confidentiality requested by participants and agreed.

## Abstract

### Background

Studies in Ghana have reported discrepancies between trends in Total Fertility Rate (TFR) and Contraceptive Prevalence Rate (CPR). Yet, there is limited empirical literature on stakeholders' perceptions on the trends in CPR and TFR in Ghana. We, therefore, examined the perceptions of key stakeholders about the documented trends in CPR and TFR in Ghana.

### Methods

We adopted an exploratory (qualitative) research design with a qualitative approach of data collection from stakeholders in Ghana, focusing on the trends of the TFR and CPR. The Consolidated Criteria for Reporting Qualitative Studies (COREQ) checklist provided additional guidance for reporting the study results. We employed the Theory of Planned Behavior (TPB) as a theoretical framework/construct to explain and predict individual changes in health behaviors resulting in trends in CPR and TFR from stakeholders' perspectives and analyzed the data using framework analysis approach.

### Results

Two main themes emerged from the data: contraceptive prevalence and total fertility ratio, with five sub-themes identified: barriers to contraception, motivations for contraception uptake, unmet need for family planning, induced abortion, and effectiveness of planning programs. Specifically, participants indicated that there is a discrepancy between the trends of CPR and TFR based on the Ghana Demographic and Health Survey, conducted between 1988 and 2014. The high unmet needs for contraceptives were attributed to CRP trends, whilst abstinence, infertility, and high demands for induced abortions were identified to impact the TFR trends significantly.

**Funding:** The authors received no specific funding for this work.

**Competing interests:** The authors have declared that no competing interests exist.

**Abbreviations:** AIDS, Acquired Immune Deficiency Syndrome; ARHR, Alliance for Reproductive Health Rights; COVID-19, Coronavirus 2019; CPR, Contraceptive Prevalence Rate; DKT, DKT International was named in honour of D.K. (Deep) Tyagi. Mr. Tyagi was the assistant commissioner of family planning in charge of public motivation and education aspects of India's family planning program until he died of cancer in 1969 in New Delhi; FP, Family Planning; GDHSs, Ghana Demographic and Health Surveys; GFPCIP, Ghana Family Planning Costed Implementation Plan; GHS, Ghana Health Service; GSS, Ghana Statistical Service; HIV, Human Immuno-deficiency Virus; ICF, Inner-City Fund; NGOs, Non-Governmental Organisations; NHIS, National Health Insurance Scheme; NPC, National Population Council; PPAG, Planned Parenthood Association of Ghana; RIPS, Regional Institute of Population Studies; TFHO, Total Family Health Organisation; TFR, Total Fertility Rate; UNFPA, United Nations Population Fund; USAID, United States Agency for International Development; WHO, World Health Organisation.

## Conclusion

The findings show that an extensive quantitative enquiry into the exact relationships between Ghana's CPR and TFR as well as the contributions of abstinence, infertility, and induced abortion are worth considering.

## Introduction

Contraceptives, also known as contraception or birth control methods, are the medicines, devices, or surgery intended to prevent pregnancy [1]. Contraceptive prevalence is the proportion of women who are currently using, or whose sexual partner is currently using, at least one method of contraception, regardless of the method being used [2]. A related concept is Contraceptive Prevalence Rate (CPR), which is defined as the percentage of women of reproductive age who are currently using at least one contraception method, or whose partner is using a contraception method, irrespective of the method [3].

CPR is commonly computed for married or in-union women between 15 and 49 years because for a given year, contraceptive prevalence measures the percentage of women of child-bearing age in union who use a form of contraception. It is a key indicator for measuring improvements in access to reproductive health service as reflected in target 3.7 of the 2030 Agenda for Sustainable Development, i.e. "By 2030, ensure universal access to sexual and reproductive health-care services, including for family planning, information and education, and the integration of reproductive health into national strategies and programs" [4].

A rise in CPR is a critical proximate indicator of inter-country variations in fertility decline [5]. It is a determining factor for population and health, predominantly access to reproductive health services that are essential for meeting several health targets, particularly those that are related to maternal and child health, HIV/AIDS, and gender equality [6–8]. Evidence from Ghana [9, 10] and elsewhere [11, 12] have shown that contraception use is affected by fertility preferences, availability and access to high-quality family planning products and services, educational attainment, social values, marital patterns among others. Other scholars have also reported the influence of socio-economic standing, knowledge about contraception, partner's preferences, cost, and level of education [13–15].

In Ghana, successive governments have made commitments toward contraceptive use, and these intensified after maternal mortality was declared a national emergency. The commitment manifests in the development of a legislative instrument to highlight Family Planning (FP) services under the National Health Insurance Scheme (NHIS) and the implementation of the Ghana Family Planning Costed Implementation Plan, 2016–2020 (GFPCIP) [16, 17]. Family Planning commodities and services are also provided free of charge across public health facilities nationwide coupled with expansion in contraceptive methods [18]. Since 2012, there has been a surge in stakeholders who intend to expand quality and access to FP services both in public and private spaces, including faith-based organizations [18].

CPR has a direct and strong relationship with the total fertility rate (TFR), and subsequently on population growth of any given country, such that ascendency in CPR is anticipated to be associated with a proportional decline in TFR [19, 20]. Conversely, the trends in Ghana's CPR and TFR appear to digress from this pattern. For instance, the 2014 Ghana Demographic and Health Survey reported that 23% of all women were using any contraceptive method, with 27% contraceptive use among currently married women and 45% among sexually active unmarried women [21]. A retrospection of CPR shows that contraceptive use among married women doubled between 1988 and 2014, thus from 13% to 27% [22]. Within the same period,

TFR only declined from 6.4 children per woman to 4.2 children per woman. In Figs 1 and 2, the trends in CPR and TFR from 1998 to 2014 are graphically illustrated.

As shown in Figure, the TFR in Ghana declined sharply from 1988 to 1998, plateaued between 1998 to 2003, dropped slightly from 4.4 to 4.0 in 2008, and rose to 4.2 in 2014.

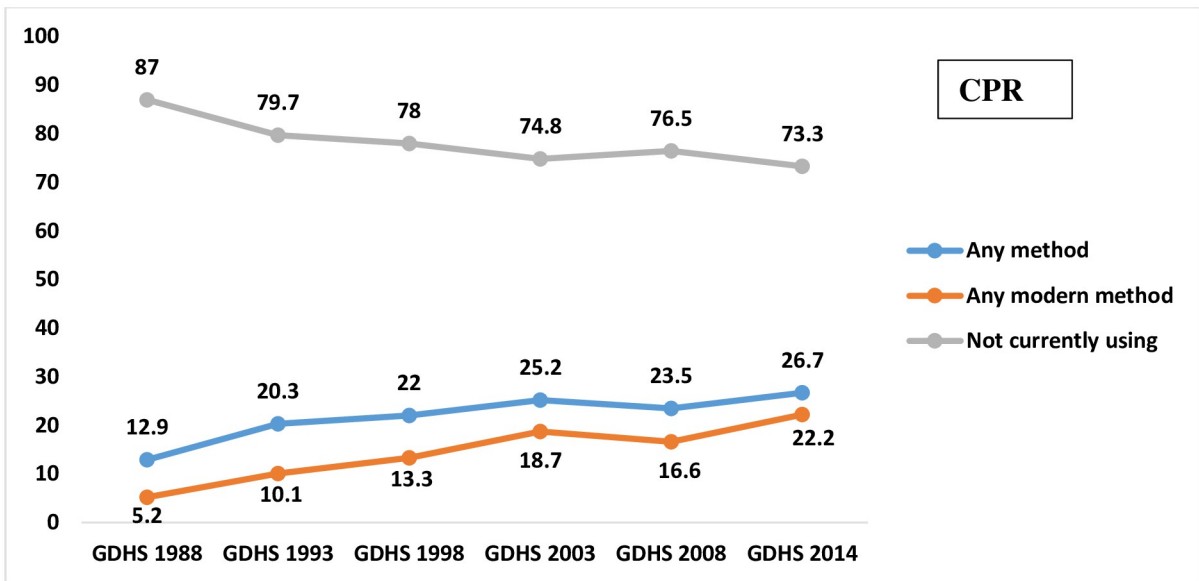

**Fig 1. Trends in contraceptive use in Ghana: 1988–2014. Source:** GDHS 1988, 1993, 1998, 2003, 2008, 2014.

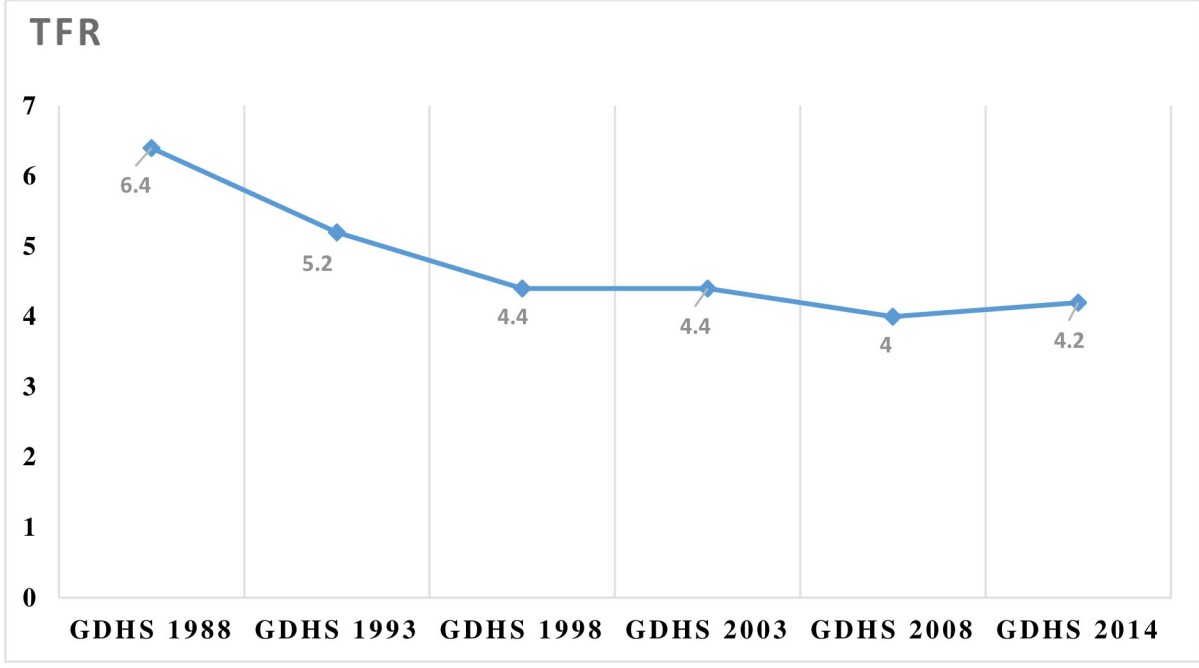

**Fig 2. Trends in Total Fertility Ratio in Ghana: 1988–2014. Source:** GDHS 1988, 1993, 1998, 2003, 2008, 2014.

Previous studies have investigated CPR and associated factors alone [23–25] whilst a few investigated TFR and its associated factors [26–28]. Despite the insights from these standalone studies, they do not give a clear understanding and appreciation of the prevailing disproportional association or relationship between CPR and TFR. The proposed study intends to bridge this knowledge gap to expand the frontiers of knowledge on key stakeholders' perceptions of the trends in Contraceptive Prevalence Rate and Total Fertility Rate in Ghana using a qualitative approach. The main objective of the study was to examine the various stakeholders' perceptions of the trends of CPR and TFR in Ghana as well as how the CPR has affected the TFR in Ghana over the years. The study findings are anticipated to extend the frontiers of knowledge to guide national family planning programs and policy directions. We employed the Theory of Planned Behavior as a theoretical framework/construct for the study to explain and predict individual changes in health Behaviors resulting in trends in CPR and TFR from stakeholders' perspectives.

## Theoretical underpinning

The Theoretical Framework driving this study was based on the Theory of Planned Behavior (TPB). The TBP suggests that individuals act rationally according to their attitudes, subjective norms, and perceived Behavioral control [29–31]. In this regard, to use contraceptive(s) to prevent pregnancy and childbirth, an individual must evaluate it positively, believe it is worthwhile, and perceive it to be within their own control. Originally, the theory helps to explain attitudes of individuals towards a Behavioral change or modification. However, in the present study, it is aiding us to engage stakeholders to offer explanation to the attitude of Ghanaian women toward contraception and how this has shaped and continues to shape the country's TFR.

The use of contraceptives involves a number of key factors, including attitude, subjective norm, and perceived behavior control. Subjective norm is based on the social context, attitude is based on results, and perceived behavior control is based on challenge. The person's attitude, subjective norms, and sense of control have an impact on their decision to use contraceptives to regulate their fertility. Stakeholders in healthcare have a vested interest in contraceptive decisions. Hence, through the lens of the TPB, we are able to explore the stakeholders' expert opinions about contraceptive use and fertility trend in Ghana, to guide policy and fertility regulation implementation strategies. Using the TPB to explain Stakeholders' Perceptions of the Trends in Contraceptive Prevalence Rate and Total Fertility Rate in Ghana is strongly supported by our empirical data, which suggest that attitudes, subjective norms, and perceived behavioral control are critical contributory factors towards fertility regulation [32].

## Methods

### Setting

**Study design.** We adopted an exploratory (qualitative) research design with a qualitative approach for data collection and analysis because from the literature reviewed the research topic is relatively a new issue hence there were no pre-existing knowledge or paradigm to study it. it. The Consolidated Criteria for Reporting Qualitative Studies (COREQ) checklist [33] provided additional guidance for reporting the study results. The choice of this integrated design aligns with the study objective to examine from the stakeholders' perspectives, their views about the associations between Ghana's CPR and the TFR within 26 years (1988 to 2014) in Ghana.

**Target institutions.** The main target population included officials of government and non-governmental organisations [NGOs] working in the area of Sexual and Reproductive

Health in Ghana. Officials from the following NGOs were targeted as their routine activities revolve around reproductive health and fertility issues: Marie Stopes Ghana, Planned Parenthood Association of Ghana (PPAG), DKT (D.K. (Deep) Tyagi) International Ghana, Ghana Health Service, Total Family Health Organisation (TFHO), United Nations Population Fund (UNFPA) and Alliance for Reproductive Health Rights (ARHR). Heads of public institutions or the Focal persons on population and related issues in these institutions were also selected to participate in the study: National Population Council (NPC) Ghana, the Ghana Health Service (GHS), The Ministry of Health (MoH), The Ministry of Women, Gender and Social Protection (MoWGSP). Besides, some academics with demonstrable evidence of working in the area of sexual and reproductive health were targeted from the following institutions: University of Ghana (specifically Regional Institute of Population Studies [RIPS] and the Department of Population, Family and Reproductive Health), University of Cape Coast (thus Department of Population and Health). For us, the participants' perceptions of CPR and TFR were valid despite the duration of exposure/involvement in contraception/sexual and reproductive health issues in Ghana and internationally.

**Selection of participants.** Participants for the study were purposively selected from the targeted institutions. This comprised the focal persons for population, sexual and reproductive health related issues as well as frontline family planning providers in identified health institutions to obtain rich and detailed information on the research topic. because we had a perception of the type of participants we wanted to interview, recruitment of participants was done until a point of saturation was reached on major issues that were outlined to be examined. In all, 15 stakeholders consented and participated in the study. It has been recommended that qualitative studies require a minimum sample size of 12 to reach data saturation [34, 35]. The institutions that participated in this study were drawn from the various well-established institutions involved in family planning activities in Ghana (see distribution in Table 1).

**Research instruments.** For data collection, the authors prepared a semi-structured interview guide. This research tool was pretested with other employees at the same specified organizations who were knowledgeable about the issues at hand but weren't the study's target participants. The feedback from the research instrument's preliminary testing was utilized to refine and finalize the tool for usage in the field.

**Data collection procedure.** In-depth interview was the main data collection method. The interviews were conducted face-to-face and via phone calls. The research team visited each of the listed organisations and scheduled appointments with the focal person for Sexual Reproductive Health and Rights for interviews to be held at a mutually agreed time and location.

**Table 1. Targeted institutions and distribution of participants.**

| Targeted Institutions & Number of Participants | NGOs in SRHR | | | | | | | Public Institutions | | | | Academia | | | Expect Opinion | Total |
|---|---|---|---|---|---|---|---|---|---|---|---|---|---|---|---|---|
| | MSG | PPAG | DKT | USAID | TFHO | UNFPA | ARHR | NPC | GHS | MoWGSP | MoH | RIPS (UG) | DPH/ UCC | SPH (UG) | FP PROVIDER | |
| **Number of Participants** | 1 | 1 | 1 | 1 | 1 | 1 | 1 | 1 | 1 | 1 | 1 | 1 | 1 | 1 | **1** | 15 |

MSG-Marie Stopes Ghana; PPAG-Planned Parenthood Association of Ghana; MoH-Ministry of Health; GHS-Ghana Health Service; TFHO-Total Family Health Organisation; UNFPA-United Nations Population Fund; ARHR- Alliance for Reproductive Health Rights, NPC-National Population Council; PIPS-Regional Institute of Population Studies; DPH/UCC- Department of Population and Health, University of Cape Coast; SPH/UG-School of Public Health, University of Ghana; DKT-D.K. (Deep) Tyagi and FP-Family Planning Provider.

Preliminary discussions on the research topic and objectives were done virtual including phone or video conferencing applications (e.g., Zoom and WhatsApp). The actual interviews were however done face-to-face and on one-on-one basis in the respective participants workplaces (Offices), thus a location where only the participant and interviewer were present to ensure optimal confidential discussions. The authors collected data with a semi-structured interview guide. The semi-structured interview guide assisted to investigate a number of issues including factors accounting for Ghana's TFR trend, the pattern of contraception use, characteristics of contraception users, and the impact of CPR on TFR in Ghana. The interviews lasted for about 45 minutes on the average. Where permitted, some of the interviews were recorded and detailed field notes were also taken by the researchers during data collection. The field work took place between July to August 2022. Data collection was done till saturation was reached.

### Research team and reflexivity

The team was made up of two experienced mixed method researchers. Both authors have executed several qualitative studies and have over a decade experience in qualitative research. We acknowledge our roles as researchers in this study because, as qualitative researchers, we were involved in the research process and our prior experiences, presumptions, judgments, and practices were probably able to have a positive influence on the research process without being biased or judgmental.

**Data analytical procedures.** The data analysis was based on the Framework analysis approach which has been used in a multitude of settings although primarily used in health care [36, 37] and the thematic analysis approach using emerging themes from the interviews. With these approaches, all audio-recorded interviews were first transcribed verbatim in the original interview language (English). Following this, all transcripts were edited to improve the flow and understanding of the text. The Framework Analysis Approach (comprising familiarisation, identification of thematic framework, indexing, charting as well as mapping and interpretation) was utilised. With this, the researchers listened to the audio tapes, read over the transcripts severally to get immersed in the data and make some preliminary notes on recurrent themes (familiarisation).

This was followed by the identification of a thematic framework whereby the research team identified emerging themes in the data. The themes or key issues subsequently became the basis of the thematic framework used in filtering and classifying the data as opined in a previous study [37]. Thirdly, at the indexing stage, we identified sections of the data/transcripts corresponding to specific themes. The fourth stage, charting, is where the specific pieces of indexed data in the third stage were arranged in charts of the themes. The final stage involved arranging the responses in themes and sub-themes for thematic analysis. All aspects of the data analysis were manually done.

**Ethical considerations.** Ethical approval was sought from the Ghana Health Service Ethics Review Committee (Protocol ID NO: GHS-ERC 009/12/21). All ethical precautions including informed consent, confidentiality, anonymity, and privacy were duly observed. Prior to the fieldwork, focal persons in the identified institutions were contacted via mobile phone and in person to solicit their consent for participating in the study. Those who verbally consented to participate in the study were then provided with the details of the study and given the chance to provide written consent. Data was collected in real time using semi-structured interview guides and no minors were involved in the study.

### Results

In Table 2, we presented the Socio-demographic characteristics of participants in the study.

**Table 2. Socio-demographic characteristics of participants.**

| Background of Participants | Years of experience in Family Planning | Gender | Age (Years | Highest educational attainment |
|---|---|---|---|---|
| Director of a family planning institution | 8 years | Female | 57 | Masters in Public Health |
| Family planning program Manager | 5 years | Female | 36 | Masters in Public Health |
| Country Representatives of an international organization | 4 years | Male | 48 | Masters in Public Health. |
| Family Planning Focal Person | 6 years | Male | 32 | Masters in Public Health |
| Frontline Family Planning Service Provider | 15 years | Female | 50 | Masters in Public Health |
| Academic from a recognised University in Ghana | 10 years | Male | 42 | PhD in Population and Health |
| Individual experts in Family planning | 20 years | Female | 65 | PhD in Population and Health |

Two major themes were identified in advance and five sub-themes were derived from the data (Fig 2).

## Contraceptive prevalence developments in Ghana

Regarding the issue of the trends in Ghana's Contraceptive Prevalence, participants expressed their views in the follow the quotes:

> *The CPR influences TFR and this happens through the control of childbirth'.*

(Participant 1)

> *'I think the contraceptive prevalence rate is actually improving, I am sure you are referring to the DHS at the time when it was quite low, I think now from the maternal health survey we are talking probably about 25% which I think we have seen a lot of improvement since the 1980s and of course, you would expect that once the contraceptive use is going high, fertility rate will be dropping equally at the same rate but unfortunately we don't see the same drop, the drop in fertility rate is quite marginable so I think that there's some work to be done around that thought.'*

(Participant 2)

## Barriers to contraceptive use

Different barriers were recounted by the research participants. The barriers identified boarded on socio-cultural issues serving as impediment to contraception uptake, All the participants held a perception that there are various barriers to contraceptive use in Ghana. Some of their views on these issues were recounted as follows:

> *'. . .Stigma, negative perceptions, ignorance, cultural values, patriarchy, and right of women on choice as well as cost of contraceptives are barriers to contraceptive use'.*

(Participant 3)

> *'Ghana is an extremely religious country, with various cultural practices that hinder contraceptive use. . .. . .lack of information about services, low knowledge about contraceptive, limited options, shyness to seek contraceptive information, access to preferred contraceptive methods, are some barriers affecting contraceptive prevalence'.*

(Participant 4)

> *'Limited number of family planning clinics coupled with myths and misconceptions around use of family planning methods affect women access to family planning services.'*

(Participant 5)

## Motivations for contraceptive use

Participants were of the view that the key motivations for contraceptive use is that contraception reduces unintended pregnancies, STI/HIV transmissions, and education of girls. Some participants explained that:

*'It's about the number of children that people want to have, the value for family size, people's understanding around limiting and spacing child birth. women desire to attain higher levels of education before childbirth, . . . . . .'*

(Participant 7)

*'. . ...if you look at the northern part of Ghana, the lack of motivation for contraceptive use is a key reason why we have very high fertility rate among people who marry at a very early age . . .. . .once they marry very early, they would have many more children than people who marry quite later in their lives. . ...'*

(Participant 8)

## Stakeholders' perceptions about Total Fertility Ratio

Participants generally agreed that while Ghana's TFR has somewhat decreased over time, there hasn't been a meaningful correlation between the TFR and the country's CPR. A rise in CPR should imply a consistent drop in TFR, as would have been predicted. Some participants specifically expressed their views in the follow the quotes:

*'There's an improvement in Ghana's Total Fertility Ratio but these trends are still not good looking too good compared to the efforts being put into family planning programs.'*

(Participant 9)

*'There is no ideal fertility rate, it's all about people's ability to take care of their children. However, the main drivers of fertility in Ghana include the desire for children, large family size, religious issues, cultural issues, gender issues, education, economic burdens.*

(Participant 10)

*'Of course we should expect that over time the fertility rate would drop, but again if you look at the total fertility rate I'm sure when you can also look at it on rural-urban divide and you can also look at northern southern divide, I'm sure you will see that we either still have very high fertilities in the rural areas compared to urban or look at the northern part and the fertility rate is much higher than you find in the southern part of the country and therefore all these put together you would expect the total fertility rate to still remain a bit high.'*

(Participant 11)

## Unmet needs for family planning

The percentage of married or women in a union of reproductive age who do not use any kind of family planning but would prefer to delay their next pregnancy (unmet need for spacing) or do not wish to have any more children (unmet need for limiting) according to the participants reflects the unmet need for contraception in Ghana. Despite the various interventions to increase contraceptive use in Ghana, participants in this study indicated that there are still

some factors leading to high unmet needs for family planning among specific populations in various communities in Ghana. Participants indicated that:

'. . .yes, there are still people in Ghana who need some type of contraceptives but can't afford the cost especially the surgical methods. . ..'

(Participant 12)

'The unmet needs for family planning are quite high whether among unmarried women or even among adolescents, in our country we still have many factors playing together that impact on the fertility rate as we would have expected.'

(Participant 13)

## Effectiveness of family planning programs

Participants were of the view that the effectiveness of family planning programs in Ghana should be a situation where family planning interventions are seen to boost social and economic growth while also enhancing health and reducing poverty. Some of the participants questioned the effectiveness of family planning programs in Ghana as follows:

'Our family planning programs is full of duplication. . .. . .. you get to the communities to find various NGOs doing the same things. . .it would have been best if there is proper coordination to ensure proper integration, humanisation and leveraging of efforts to ensure program effectiveness.'

(Participant 14)

'I don't think our national family programs are being effective. . .. . .. we still have very high unmet needs for family planning, we still have some socio-cultural beliefs, around, even in urban cities, people still value larger families and also, we still have the male dominance in family planning decision making. . .. . .. . .. some women who need family planning services are been prevented by their husbands who still value higher number of children.'

(Participant 15)

## Perceived relationships between TFR and CPR

Commenting on the observed trends in the TFR and CPR as indicated in Figs 2 and 3, some participants shared their views as follows:

'There are speculations that the declined CPR in Ghana is being influenced by induced abortion, abstinence or infertility but There is no empirical evidence to prove this. Because induced abortion is not easily accessible in Ghana, it cannot be explicitly said that induced abortion is being used as a family planning/birth control option in Ghana'.

(Participant 1)

'Logically, CPR must influence TFR because as contraceptives use increases, fertility is expected to decline proportionately. . .. However, the relationship between the CPR and TFR are not proportional and not consistent'

(Participant 2)

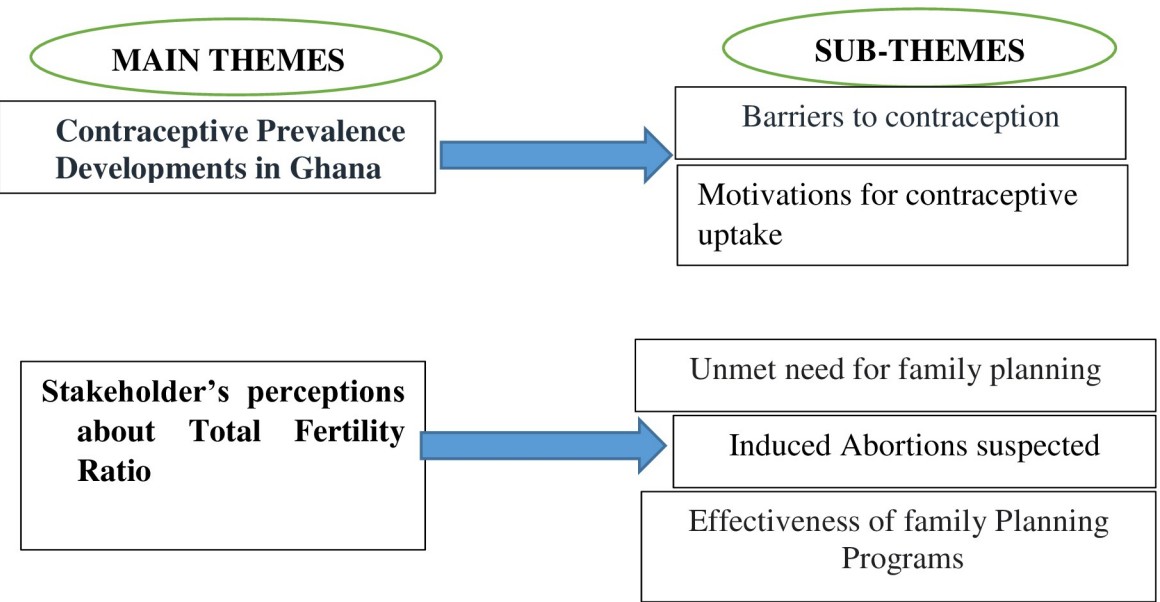

**Fig 3. Themes and subthemes that emerged from the interviews.** Regarding the analytical approach and preliminary findings, results were shared with the participants and they offered input. Participant quotations were then presented to illustrate the themes/findings.

*'I would* say *there should be an inverse relationship between the two such that if your CPR is increasing which means you are getting many more women either deciding to limit or appropriately space the number of children that they have, we should also expect the TFR to be reducing. So, I will say there should be an inverse relationship between the two under ideal circumstances. But of course, as I indicated earlier there are a number of factors that play into that you can have your CPR going up but you would not necessarily find the same level of drop in your TFR'.*

(Participant 3)

*'. . ..no, it's not proportional, we've seen a huge growth in the CPR, if you look at it from the last DHS that was done before the 2010 one, the CPR was probably the teens, now I'm seeing we've gone above 25%, that's probably almost half but if you look at the TFR, I don't think you found the same appropriate level of drop in the TFR, probably due to other factors including induced abortion'.*

(Participant 4)

*'I think although abstinence, infertility and induced abortion are the key drivers of the discrepancy between the CPR and TFR in Ghana. . .the contribution of Induced abortion to this is very huge. . . . . .this is evident from the many abortion clinics available in Ghana and the huge number of clients patronising abortion services daily in these facilities which excludes the many more who self-induce abortion or use the 'back street services'*

(Participant 5)

## Induced abortion role

The role of induced abortions in Ghana's CPR and TFR has been a key issue of contention. Although there was no empirical evidence to back the claim, 12 out of the 15 participants

suspected induced abortion as a major contributing factor to the discrepancy between CPR and TFR and expressed their views as follows:

> *'Induced abortion is readily available in most registered hospitals in Ghana, yet there is no real data on abortion because it is under-reported. . . . . . . .because there is no data, currently it cannot be concluded that induced abortion is being used as family planning which reflects on the levels of the TFR and CPR'.*

(Participant 6)

> *'I think induced abortion might be a contributory factor but I'm not sure it is the main driver. . .. of course if in a year you have close to hundred thousand pregnancies being termi-nated in our facilities, cumulatively, over time it will have an impact but that is so minor, that's why I'm telling you you're looking at hundred thousand verses even if you look at our reproductive age group, you are talking about nearly 35 to 45% of the population being in their productive age, if you take 45% that's almost half of the population and we are talking about 15million, 100,000 against 15million is quite minute to be able to see any significant impact on our CPR and TFR'.*

(Participant 7)

## Discussion

In this study, we first generally examined the views of Sexual and Reproductive Health and Rights stakeholders on how the Contraceptive Prevalence Rate (CPR) has affected the Total Fer-tility Ratio (TFR) of Ghana over the years and situated the study within the Theory of Planned Behavior (TPB) [29–31]. We then specifically examined the participants' views on the main study objective by a tailored discussion on effectiveness of family planning programs, the role of abortion in TFR trends and unmet need for family planning in Ghana. We observed that, despite the efforts being made by stakeholders to improve the CPR in Ghana, the barriers to contracep-tion seem to have far outweighed the motivations for contraception as this was also observed in a previous similar study [38]. The theoretical framework (i.e., TPB) [29–31] in which we situated this study provides a better lens for interpreting our findings. For instance, we observed that, issues relating to participants' perceptions about CPR and TFR associations in Ghana were dis-cussed within the context of health care seeking Behavior and the perceived relevance of contra-ception to the health and well-being of the individuals who opted to use contraceptives.

Some participants were more inclined towards relating CPR and TFR to human Behaviors; believing that contraception uptake is a Behavioral issue and influenced by human perception about the benefits that one could accrue from contraception to achieving optimal reproductive intentions including decisions on fertility. As established in the literature, several factors affect the decision to use contraception and the type to use [34, 36–38] The findings suggest that interventions to enhance contraception uptake must be multifaceted in order to offset as many bottlenecks to contraception use as possible. For instance, in addition to advertisements, it is also expedient to ensure that contraceptive services are readily available to all reproductive-aged women irrespective of geographical location and wealth status. Ensuring this can contrib-ute positively towards Ghana's prospects of achieving the seventh target of the third Sustain-able Development Goal [38–40].

Some stakeholders held the perception that the reduction in TFR is slow and do not com-mensurate the CPR. This observation by the stakeholders is plausibly due the fact that not all contraceptive users intend to halt childbearing, as some people use contraception for birth

spacing [41, 42]. Besides, it is worthy of mention that fertility reduction is partly Behavioral and cannot be achieved mechanically. Hence, reduction in fertility is a gradual process premised on individuals' reflections, preferences and critical evaluation of the merits and demerits. In consonance with our study, some previous studies in Ghana [43, 44], realised that contraception use only corresponds partially with fertility reduction.

Participants indicated several factors that have been affecting contraceptive use in Ghana at their institutional levels as well as based on work experience in family planning. These were also observed by a similar previous study organised under the following four main headings: household and individual, community, regional and national [43, 44]. These factors influence contraceptive use through their effects on the demand for and supply of births. For example, at the national level, social policy, the economic situation, government and donor support for family planning, and family planning implementation play a crucial role in promoting contraceptive use. Collectively these factors at the national, regional, community, household, and individual levels determine rates of contraceptive use in a given country. From these perspectives, it can be deduced that in some communities, unmet needs for family planning could multifaceted including ineffective systems, inadequate supervision as well as monitoring of program/project deliverables at the community level where there is the most need for family planning services.

## Limitations

This study is limited by the type of design and the sample size which prevents the findings from being generalised. Despite this limitation, the study findings may be useful as a pilot to guide future studies of wider scope.

## Conclusions

Our qualitative inquiry into stakeholders' perceptions of the trends in contraceptive prevalence rate and total fertility rate in Ghana provides some relevant information that could be further explored. The Theory of Planned Behavior (TPB) used to drive the study has been useful in identifying a number of key factors including attitude, subjective norm, and perceived Behavior control which relate to stakeholders' perceptions about Ghana's TFR and CPR. Subjective norm is based on the social context, attitude is based on results, and perceived Behavior control is based on challenge. We argue that a person's attitude, subjective norms, and sense of control have an impact on their decision to use contraceptives to regulate their fertility and also facilitating the understanding of possible pathway through which Behavioral issues inform CPR and TFR in Ghana. In this regard, there might be a need to consider incorporating collecting data on induced abortion in future GDHS to understand the exact contributions of induced abortion to the TFR. Although such an enquiry could be a challenging task due to the high abortion stigma, which might hinder realistic data collection, an innovative design coupled with intensifying public education on abortion and the relevance of such data might yield the desired results. As highlighted by the TPB, both enabling and obstructing factors must be prioritised in the national security as well as contraception up initiatives/strategies to enhance chances of achieving a linear correlation between CPR and TFR which is the expected and desired result.

## Supporting information

**S1 Checklist. COREQ (COnsolidated criteria for REporting Qualitative research) checklist.**
(DOC)

## Acknowledgments

We are grateful to all those who provided information and direction for this paper.

## Author Contributions

**Conceptualization:** Fred Yao Gbagbo.

**Data curation:** Fred Yao Gbagbo, Edward Kwabena Ameyaw.

**Formal analysis:** Fred Yao Gbagbo, Edward Kwabena Ameyaw.

**Methodology:** Fred Yao Gbagbo, Edward Kwabena Ameyaw.

**Supervision:** Fred Yao Gbagbo.

**Validation:** Fred Yao Gbagbo, Edward Kwabena Ameyaw.

**Writing – original draft:** Fred Yao Gbagbo.

**Writing – review & editing:** Fred Yao Gbagbo, Edward Kwabena Ameyaw.

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
