## [Decision Letter · Decision Letter 0]

6 Apr 2023

PONE-D-22-27258Qualitative Enquiry into Stakeholders’ Perceptions of the Trends in Contraceptive Prevalence Rate and Total Fertility Rate in Ghana through the lens of the Health Belief Model 1Fred Yao Gbagbo and 2Edward Kwabena AmeyawPLOS ONE

Dear Dr. Gbagbo,

Thank you for submitting your manuscript to PLOS ONE. After careful consideration, we feel that it has merit but does not fully meet PLOS ONE’s publication criteria as it currently stands. Therefore, we invite you to submit a revised version of the manuscript that addresses the points raised during the review process.

 The Reviewers have given constructive comments bothering on study design, methodology, analysis and presentation of results. Authors are encouraged to all address all comments raised by the reviewers which will ultimately improve the manuscript to the standard expected. The manuscript will benefit from a major revision of the various sections as commented. 

We look forward to receiving your revised manuscript.

Kind regards,

Bismark Dwumfour-Asare, MSc, PhD

Academic Editor

PLOS ONE

Journal Requirements:

2. During your revisions, please note that a simple title correction is required: Please remove the author names "1Fred Yao Gbagbo and 2Edward Kwabena Ameyaw" from the title on the online submission information.

Additional Editor Comments (if provided):

Our reviewers have done a thorough assessment of the manuscript and have given constructive comments and suggestions for the improvement of the manuscript to meet the standard required by the journal PLOS ONE before further consideration. The manuscript could have a potential for publication if the reviewers queries are addressed adequately moving forward.

Reviewers' comments:

Reviewer's Responses to Questions

**Comments to the Author**

1. Is the manuscript technically sound, and do the data support the conclusions?

Reviewer #1: Partly

Reviewer #2: Yes

Reviewer #3: Partly

2. Has the statistical analysis been performed appropriately and rigorously? 

Reviewer #1: Yes

Reviewer #2: N/A

Reviewer #3: N/A

3. Have the authors made all data underlying the findings in their manuscript fully available?

Reviewer #1: No

Reviewer #2: Yes

Reviewer #3: No

4. Is the manuscript presented in an intelligible fashion and written in standard English?

Reviewer #1: Yes

Reviewer #2: Yes

Reviewer #3: Yes

5. Review Comments to the Author

Reviewer #1: GENERAL COMMENTS

The authors did a good job. However, the paper talks about a concept that is already known, that is Contraceptive Prevalence Rate (CPR) and Total Fertility Rate (TFR) are inversely proportional; that is, an increase in CPR implies a decrease in TFR and vice versa. The paper also talks about another known fact which is a percentage increase in one does not always imply a reduction in the other by the same percentage.

Based on the results obtained, the Health Belief Model may not have been the best approach.

The points raised in the conclusion for future work is what the authors should have worked on. They have missed something crucial.

The authors said some restrictions exist in accessing data for the study. They did not state where data can be found.

If the paper is to be published, it needs major revision.

Specific comments

1. The paper has a number of outstanding typographical/grammatical issues. A few examples are stated below:

i. Misplaced/wrongly used punctuation

a. Page 3 line2: “?” after service

b. Page 11: inverted comma to close quote by first participant is missing

c. Page12: inverted comma to close quote by participant is missing

d. Page 13: inverted comma to close quote by participant 10 is missing

e. Page 16: Quote by participants 5 & 7 is not in inverted commas

ii. Both British and American English used

a. Page 6:

b. Page 14: … Family Planning Programs …

c. Page 14: quote by participant 14 … programme …

d. Page 9 line 6: … familiarization

e. Page 9 line 9: … familiarization …

iii. Present and past tense have been mixed up in some instances

iv. Misnomers in some instances

a. “Tenant” instead of “tenet”

v. Abbreviations being used without stating it in full first

a. page 7 line 5: what does the “DKT” stand for?

Reviewer #2: This important study attempts to provide an expert view of recent trends in CPR and TFR using qualitative research methodology. The study is timely and if published will provide relevant basis for sexual reproductive health interventions in Ghana and elsewhere. However, major issues have been identified and where possible suggestions to improve the quality of the manuscript have been provided to the authors.

Below are my comments for the authors.

1. Theoretical underpinning

The health belief model is not an appropriate theory for this study since it focuses on individual level health behaviours. In addition, the findings do not emphasize the constructs of HBM. A relevant community/population level theory for this study could be “critical realism” – a theoretical framework that combines a realist ontology with an interpretivist epistemology. Thus, the view that unobservable structures cause observable events (e.g trends in CPR) and that these observations can be understood if we understand the mechanisms that cause them. The authors are at liberty to consider other appropriate theories.

2. Study design

(i) The authors report an integrated descriptive and exploratory research design with a qualitative approach. However, the element of integration of descriptive and exploratory designs is neither clear nor appropriate. This work is solely exploratory (qualitative) so should be reported as such.

(ii) Many COREQ items have not been reported or are not clear from the reporting. Items 1-5, 7,8,13,23,27,28, and 32 seems unreported. When a reporting checklist (e.g. COREQ, SRQR, PRISMA etc..) has been used, it is always helpful to attach a list of all checklist items and where they have been reported in the manuscript.

3. Data Analytical procedures

Since the framework analysis approach is a specialized form of thematic analysis, it should be sufficient for the authors to only report the framework analysis approach.

4. Results

(i) Contraceptive Prevalence and Total Fertility Ratio cannot be considered as themes since they are the main problems that were investigated. If the authors wish to consider them as themes, then they should be reported as themes developed in advance of data collection (deductive) and not as emerging from the data (inductive) [see COREQ item 26, derivation of themes].

(ii) The authors have not provided a detailed description of the subthemes. What is seen is a presentation of several quotes with very scanty description of the findings that reflect authors’ synthesis of the data. For instance on “Barriers to contraceptive use”, the authors mention that “participants held a perception that there are various barriers to contraceptive use…”. However, they failed to detail what these main barriers were (as revealed through their analysis of the data). The authors have to develop minor themes for the barriers or state them in their findings before supporting them with quotes. Quotes are not to be used as substitutes for the research findings. Kindly note that this comment applies to all the themes reported in the findings and not just the barriers to contraceptive use.

5. Discussion

Present a detailed explanation of your findings, clearly situating them in the context of what is already known. Authors’ should try to compare key finding to existing knowledge and provide explanation for findings and not merely restate them.

5. Other comments

Kindly proofread the manuscript to correct any grammatical errors.

Reviewer #3: Thanks for the opportunity to review the manuscript titled: “Qualitative Enquiry into Stakeholders’ Perceptions of the Trends in Contraceptive Prevalence Rate and Total Fertility Rate in Ghana through the lens of the Health Belief Model”. The manuscript is of interest and falls within the scope of the journal. However, I have the following comments to strengthen the current version of the manuscript. Based on the comments that I have raised, I suggest a MAJOR REVISION.

TITLE

1. “Qualitative Enquiry into Stakeholders’ Perceptions of the Trends in Contraceptive Prevalence Rate and Total Fertility Rate in Ghana through the lens of the Health Belief Model”. This title looks too long and will benefit from modifications. I proposed the following title: “Stakeholders’ Perceptions of the Trends in Contraceptive Prevalence Rate and Total Fertility Rate in Ghana”. In this way, you would still have the opportunity to comment on the health belief model in your work. Also, readers will have the opportunity to read the abstract so they would be able to know that this study is qualitative.

ABSTRACT

Background

2. “Studies in Ghana have reported discrepancies between trends of Total Fertility Rate (TFR) and Contraceptive Prevalence Rate (CPR) as also reflected in the various Ghana Demographic and Health Survey (GDHS) reports”. You can modify this statement as: “Studies in Ghana have reported discrepancies between trends of Total Fertility Rate (TFR) and Contraceptive Prevalence Rate (CPR)”. After that, try to indicate the research gap calling for the study and the rationale of the study. Apart from that, in the background section, let it be clear why you employed the health belief model as a theoretical framework/construct for the study.

Methods

3. You have to comment on the sample size and the sampling techniques in the methods section. In the methods section of the abstract, talk about how the health belief model influenced the design of your methods. I am sure a bit of information on these issues will do. It will also help readers to have a fair idea about how the health belief model was employed in the study.

Results

4. Since you employed the health belief model, you have to organize your results based on the constructs of the health belief model. Per what you have currently done, it shows that you did not employ any model to drive your study. These issues should be fixed.

Conclusion

5. In the conclusion section of the study, you have to tie the findings to the health belief model.

MAIN WORK

Introduction

6. Page 3, “CPR is commonly computed for married or in-union women between 15 and 49 years”. Are you able to explain further why this age range is mostly considered?

7. Page 3, “It is a key indicator for measuring improvements in access to reproductive health service? I am not sure you need the question mark (?)…

Theoretical frameworks

8. Here, comment on some of the studies on contraceptives that have employed the health belief model. Also, talk about how you are going to employ the health belief model in your work.

9. “This model is a behaviour change framework that has been and is still widely used throughout the world as a theoretical model to guide health promotion programs. It is used to explain and predict individual changes in health behaviours. Based on our assumptions that contraception and fertility are informed by an individual’s reproductive intentions, the (HBM) aligns well with our study design as it defines the key factors that influence health behaviours as an individual's perceived threat to sickness or disease (perceived susceptibility), the belief of consequence (perceived severity), potential positive benefits of action (perceived benefits), perceived barriers to action, exposure to factors that prompt action (cues to action), and confidence in the ability to succeed (self-efficacy)”. These statements should be cited. Read about the following scholars and cite them in your study.

Rosenstock, I M. (1974a). Historical Origins of the Health Belief Model. Health Education Monographs Winter, 2(4), 328-335

Rosenstock, I M. (1974b). The Health Belief Model and Preventive Health Behavior. Health Education Monographs Winter, 2(4), 354-386

Rosenstock, I M. (1988). Enhancing Patient Compliance with Health Recommendations. Journal of Pediatric Healthcare, 2, 67-72.

Methods

Design

10. “We adopted an integrated descriptive and exploratory research design with a qualitative approach for data collection and analysis”. Are you able to expand this statement by justifying why descriptive and exploratory research design? This will give readers better perspectives on why descriptive and exploratory research design was used.

Target institutions

11. “The main target population included officials of government and non-governmental organisations [NGOs] working in the area of Sexual and Reproductive Health in Ghana’. Apart from selecting the NGOs based on their area of focus, what other criteria did you employ? E.g. Based on the number of years that they have been into sexual and reproductive health and other issues? Please, do clarify as this may have impacted on their perceptions of contraception/sexual and reproductive health ….

12. “Marie Stopes Ghana, Planned Parenthood Association of Ghana (PPAG), DKT International Ghana, Ghana Health Service, Total Family Health Organisation (TFHO), United Nations Population Fund (UNFPA) and Alliance for Reproductive Health Rights (ARHR)”. How many years have these NGOS being in operation?

13. “Heads of public institutions or the Focal persons on population and related issues in these institutions were also selected to participate in the study: National Population Council (NPC) Ghana, the Ghana Health Service (GHS), The Ministry of Health (MoH), The Ministry of Women, Gender and Social Protection (MoWGSP)”. How many years have they been working with these institutions? Any information on that? This information could influence the quality of the information these stakeholders will provide.

14. “Besides, some academics with demonstrable evidence of working in the area of sexual and reproductive health were targeted from the following institutions: University of Ghana (specifically 8 Regional Institute of Population Studies [RIPS] and the Department of Population, Family and Reproductive Health), University of Cape Coast (thus Department of Population and Health”. How many years have they been in academia?

Participants selection

15. “As the emphasis of qualitative research is not on large sample sizes, recruitment of participants was done until a point of saturation was reached on major issues that were outlined to be examined”. I don’t seem to agree with this statement because you already have a pre-determined categories of people you were going to gather data from per what you have highlighted in your target institutions. Hence, you need to reconsider your reason. I however, agree with you on the basis that a sample size of 12 is good enough for a qualitative study since the emphasis is not on generalization.

16. “In all, 15 stakeholders consented and participated in the study”. Did they consent orally or written and why? Such information may help readers…

17. “Source: Authors’ Construct 2022”. Remove this information, you don’t need it especially in publications.

Data Collection Procedure

18. Were interviews conducted in English? if yes, why? What were some of the trustworthiness/rigor issues such as credibility, transparency, dependability, and confirmability that were considered in your study? Include information on how you applied these issues in your work. One key issue is that you are yet to capture information on how you design your data collection instrument to reflect the domains/dimensions of the health belief model… So far, the above information is missing in your work…Further, as part of the data collection procedure, you should also be able to provide information on research team and reflexivity. This will also include your positionality and intersectionality issues. All these information can be captured as part of the research team and reflexivity… What is your position in this study? Are you entering this research as an outsider/insider or what? How does your outsider/insider perspective shape the outcome of your study or influenced how you interacted with the participants during the field work?

19. “Detailed field notes were taken by a dedicated note taker during interviews”. But I guess you said it was done via phone, right? If that is the case, did you put it on loudspeaker(because if it was not on loudspeaker, your note taker cannot hear the responses of the participants)? If yes, then you should explain how you did not breach the issue of confidentiality and privacy? Also, did you allow the note taker to sign any confidentiality form? If yes, you need to include this information in the work? If no, why? Did you give training to the note taker? What was the training about? How many days for the training? These issues should be commented in the work.

20. “Data collection lasted for two months”. Just stating data collection lasted for two months may not be enough, just be specific? For instance, January 2022-March 2022. This will make things clearer for everyone who is reading the work.

Data Analytical procedure

21. “With these approaches, all audio-recorded interviews were first transcribed verbatim in the original interview language (English)”. This contradicts with your earlier statement that: “ Detailed field notes were taken by a dedicated note taker during interviews”. Please, recheck…

Results

22. Terrific! This was what I was talking about in the methods section under target institutions . I mean the years of experience in family planning. You can add it now…

23. “Table 2: Socio-demographic characteristics of participants”, just providing information on the years of experience in family planning alone cannot constitute socio-demographic information of the participants. Do you have data on their gender, age, education level etc.? If yes, add them especially the gender?

24. I suggest you present the results based on the domains of the health belief model. Also, the authors have provided quotations without any detailed write-up. The quotations should be well explained.

Discussion

25. The discussion is quite okay. The authors should well connect their findings to the health belief model. Also, the authors should be strengthened the implication part. For this reason, a section focusing on policy, practice and research implications should be devoted for that.

6. PLOS authors have the option to publish the peer review history of their article (what does this mean?). If published, this will include your full peer review and any attached files.

Reviewer #1: No

Reviewer #2: No

Reviewer #3: No

---

## [Decision Letter · Decision Letter 1]

7 Jun 2023

PONE-D-22-27258R1Stakeholders’ Perceptions of the Trends in Contraceptive Prevalence Rate and Total Fertility Rate in GhanaPLOS ONE

Dear Dr. Gbagbo,

Thank you for submitting your manuscript to PLOS ONE. After careful consideration, we feel that it has merit but does not fully meet PLOS ONE’s publication criteria as it currently stands. Therefore, we invite you to submit a revised version of the manuscript that addresses the points raised during the review process.

ACADEMIC EDITOR:

The Reviewers' concerns raised are critical and must be resolved by the authors. The study is purported to be based on a theoretical framework which is under query - for not being fully considered. Authors therefore need to address this valid queries raised because the HBM is not fully satisfied in terms of results/findings supporting the six construct areas except in one area - perceived barriers. It must therefore be clear on this and if there should be adequate modifications to the presentation to consider/acknowledge any such significant limitations, authors should be able to do such. 

We look forward to receiving your revised manuscript.

Kind regards,

Bismark Dwumfour-Asare, MSc, PhD

Academic Editor

PLOS ONE

Journal Requirements:

Additional Editor Comments (if provided):

Thank you authors for accepting to review the manuscript. The reviewers have done the second round of review and have raised a fundamental issue that needs to be resolved before the manuscript can be allowed to go through the next stage of the process. Please, kindly take time and address the concern appropriately in the paper and in your response.

Reviewer's Comments:

1. To a large extent, you have addressed the issues I raised earlier. However, one outstanding issue is the concern about the appropriateness of your theoretical framework to which you provided the quoted response below:

Noted and thank you. However, we think the tenets of the HBM provides us with the best explanation for our paper. Estimating CPR and TFR is usually first done at the individual level of health behaviors before a population estimation is done. We adopted the HBM because our paper focused more on estimates at the individual level (CPR &TFR) of which the aggregated effects became a key finding to be explored later in-depth where we will probably consider the theories being suggested. Additionally, the mechanisms that cause the trends in the CPR and TFR were not the core focus of this qualitative study.

A few issues in your response need considering and I think critically considering your theory will greatly improve your manuscript:

a) CPR & TFR are not estimated or cannot be estimated at the individual level since they are population health indicators. Rates are measures of the frequency with which an event occurs in a defined ‘population’ over a specified period of time and are not individual health measures.

b) In the last statement of your response, you make the claim that your study did not focus on examining the mechanisms that cause the trends in CPR & TFR. I do not think that is accurate since your main objective of the study was to examine the various stakeholders’ perceptions of the trends of CPR and TFR in Ghana. You sought their perceptions on why (the mechanisms or underlining reasons) for the observed trends. If this was not what you sought to do, then it is not clear at this stage what you intended to do.

2. In your introduction you make the following claim: We employed the Health Belief Model as a theoretical framework/construct for the study to ‘explain’ and ‘predict’ individual changes in health behaviours resulting in trends in CPR and TFR from stakeholders’ perspectives.

This was not done anywhere in your manuscript. I once again suggest that your critically consider changing your theoretical framework. If you used the HBM then it is expected that at the minimum, your results section is organized to contain the constructs of the theory. So far, this is not done. There is no theme/data on five of the six constructs of HBM in your results; perceived severity, perceived susceptibility, perceived benefits, ‘perceived barriers’, cues to action, and self-efficacy. The only construct you considered is perceived barriers. To be honest with you, I do not see how you could justify your reliance on HBM for this study because clearly your data is not explained by the constructs of the theory.

Reviewers' comments:

Reviewer's Responses to Questions

**Comments to the Author**

1. If the authors have adequately addressed your comments raised in a previous round of review and you feel that this manuscript is now acceptable for publication, you may indicate that here to bypass the “Comments to the Author” section, enter your conflict of interest statement in the “Confidential to Editor” section, and submit your "Accept" recommendation.

Reviewer #2: (No Response)

Reviewer #3: All comments have been addressed

2. Is the manuscript technically sound, and do the data support the conclusions?

Reviewer #2: Partly

Reviewer #3: Yes

3. Has the statistical analysis been performed appropriately and rigorously? 

Reviewer #2: N/A

Reviewer #3: N/A

4. Have the authors made all data underlying the findings in their manuscript fully available?

Reviewer #2: No

Reviewer #3: Yes

5. Is the manuscript presented in an intelligible fashion and written in standard English?

Reviewer #2: Yes

Reviewer #3: Yes

6. Review Comments to the Author

Reviewer #2: COMMENTS TO AUTHORS

1. To a large extent, you have addressed the issues I raised earlier. However, one outstanding issue is the concern about the appropriateness of your theoretical framework to which you provided the quoted response below:

Noted and thank you. However, we think the tenets of the HBM provides us with the best explanation for our paper. Estimating CPR and TFR is usually first done at the individual level of health behaviors before a population estimation is done. We adopted the HBM because our paper focused more on estimates at the individual level (CPR &TFR) of which the aggregated effects became a key finding to be explored later in-depth where we will probably consider the theories being suggested. Additionally, the mechanisms that cause the trends in the CPR and TFR were not the core focus of this qualitative study.

A few issues in your response need considering and I think critically considering your theory will greatly improve your manuscript:

a) CPR & TFR are not estimated or cannot be estimated at the individual level since they are population health indicators. Rates are measures of the frequency with which an event occurs in a defined ‘population’ over a specified period of time and are not individual health measures.

b) In the last statement of your response, you make the claim that your study did not focus on examining the mechanisms that cause the trends in CPR & TFR. I do not think that is accurate since your main objective of the study was to examine the various stakeholders’ perceptions of the trends of CPR and TFR in Ghana. You sought their perceptions on why (the mechanisms or underlining reasons) for the observed trends. If this was not what you sought to do, then it is not clear at this stage what you intended to do.

2. In your introduction you make the following claim: We employed the Health Belief Model as a theoretical framework/construct for the study to ‘explain’ and ‘predict’ individual changes in health behaviours resulting in trends in CPR and TFR from stakeholders’ perspectives.

This was not done anywhere in your manuscript. I once again suggest that your critically consider changing your theoretical framework. If you used the HBM then it is expected that at the minimum, your results section is organized to contain the constructs of the theory. So far, this is not done. There is no theme/data on five of the six constructs of HBM in your results; perceived severity, perceived susceptibility, perceived benefits, ‘perceived barriers’, cues to action, and self-efficacy. The only construct you considered is perceived barriers. To be honest with you, I do not see how you could justify your reliance on HBM for this study because clearly your data is not explained by the constructs of the theory.

Reviewer #3: Thanks for the opportunity to review the paper titled: "Stakeholders’ Perceptions of the Trends in Contraceptive Prevalence Rate and Total Fertility Rate in Ghana". I wish to say that the paper has really improved and is ready for publication in PLOS ONE. CONGRATULATIONS!

7. PLOS authors have the option to publish the peer review history of their article (what does this mean?). If published, this will include your full peer review and any attached files.

Reviewer #2: No

Reviewer #3: **Yes: **Williams Agyemang-Duah

---

## [Author Response · Author response to Decision Letter 1]

13 Jun 2023

Has been submitted as an attachment

---

## [Decision Letter · Decision Letter 2]

7 Jul 2023

Stakeholders’ Perceptions of the Trends in Contraceptive Prevalence Rate and Total Fertility Rate in Ghana

PONE-D-22-27258R2

Dear Dr. Gbagbo,

We’re pleased to inform you that your manuscript has been judged scientifically suitable for publication and will be formally accepted for publication once it meets all outstanding technical requirements.

Kind regards,

Bismark Dwumfour-Asare, MSc, PhD

Academic Editor

PLOS ONE

Additional Editor Comments (optional):

Congratulations to the authors for the hard work done on this manuscript to meet the standard of this journal.

We encourage authors to pay attention to few minor issues that need to be corrected - raised here and later by the next steps (copy editors).

These issues must be addressed immediately:

- Figures and Tables must be formatted appropriately.

- Figures should be well labeled especially the axes (vertical & horizontal) to give meaning to the readings.

Reviewers' comments:

Reviewer's Responses to Questions

**Comments to the Author**

1. If the authors have adequately addressed your comments raised in a previous round of review and you feel that this manuscript is now acceptable for publication, you may indicate that here to bypass the “Comments to the Author” section, enter your conflict of interest statement in the “Confidential to Editor” section, and submit your "Accept" recommendation.

Reviewer #2: All comments have been addressed

2. Is the manuscript technically sound, and do the data support the conclusions?

Reviewer #2: Partly

3. Has the statistical analysis been performed appropriately and rigorously? 

Reviewer #2: N/A

4. Have the authors made all data underlying the findings in their manuscript fully available?

Reviewer #2: Yes

5. Is the manuscript presented in an intelligible fashion and written in standard English?

Reviewer #2: Yes

6. Review Comments to the Author

Reviewer #2: (No Response)

7. PLOS authors have the option to publish the peer review history of their article (what does this mean?). If published, this will include your full peer review and any attached files.

Reviewer #2: No

---

## [Editor Report · Acceptance letter]

13 Jul 2023

PONE-D-22-27258R2 

 Stakeholders’ Perceptions of the Trends in Contraceptive Prevalence Rate and Total Fertility Rate in Ghana 

Dear Dr. Gbagbo:

I'm pleased to inform you that your manuscript has been deemed suitable for publication in PLOS ONE. Congratulations! Your manuscript is now with our production department. 

Kind regards, 

on behalf of

Prof. Bismark Dwumfour-Asare 

Academic Editor

PLOS ONE